# Dynamically Tunable Light Absorbers as Color Filters Based on Electrowetting Technology

**DOI:** 10.3390/nano9010070

**Published:** 2019-01-06

**Authors:** Jun Wu, Yaqiong Du, Jun Xia, Tong Zhang, Wei Lei, Baoping Wang

**Affiliations:** Joint International Research Laboratory of Information Display and Visualization, School of Electronic Science and Engineering, Southeast University, Nanjing 210096, China; 220171415@seu.edu.cn (Y.D.); xiajun@seu.edu.cn (J.X.); tzhang@seu.edu.cn (T.Z.); lw@seu.edu.cn (W.L.); wbp@seu.edu.cn (B.W.)

**Keywords:** electrowetting, tunable absorbers, subwavelength metal grating, plasmon resonance

## Abstract

A device that uses the electrowetting fluid manipulation technology to realize the reversible and dynamical modulation of the local surface plasmon resonance is invented. By varying the electrowetting voltage, the distribution of fluids media surrounding the grating structure get changed accordingly, causing the modulation of the plasmonic resonance peak. The simulation results indicated that three primary colors, that are cyan, magenta and yellow (CMY) can be respectively reflected through selecting suitable structural parameters. More importantly, for the first time, the invented fluid-based devices have exhibited fine-tuning characteristics for each primary color. Finally, the device has been proved to have a large color gamut range in the Commission International De L’E’clairage (CIE) 1931 color space.

## 1. Introduction

Artificial micro-nano structures can produce various singular optical properties and phenomena not available in natural materials [1]. Photonics devices based on the artificial micro-nano structures have broad application prospects in many fields such as display [2], detection [3], imaging [4], remote sensing [5] and printing [6] because of the advantages such as designable optical properties, excellent process compatibility, and outstanding integration [7,8,9]. Benefits from the development of micromachining technology, that is the emergence of nanoprocessing methods, the feasible working band of photonic devices has extended from infrared to visible light, based on which new principles and new methods are expected to be developed for micro-nano characterization techniques and micro-nano photonic functional devices.

In recent decades, researchers have conducted extensive research on various types of micro-nano-structured photonic devices and have obtained many novel and important research results [10,11,12,13]. Color filter is one kind of common optical component that selectively reflects or transmits a specific wavelength in the visible light band and exhibits different colors. Color filters with red, green and blue primary colors are widely used in liquid crystal display, optical communication, sensor detection, imaging and a wide range of other fields [14,15]. The color filters based on periodic nanostructures have a small absorption rate of incident light, and meanwhile, the performance parameters of which, such as the central spectral position and bandwidth can be well adjusted by the structural parameters. Especially, the one based on the periodic grating structure can be easily fabricated by photolithography or nanoimprint process, which has not only low processing cost but also superior reflection and transmission properties, has attracted numerous interests [16,17,18,19,20,21,22]. Researchers found that the grating-based nanophotonic devices exhibit singular optical effects such as guided mode resonance and surface plasmon resonance. Furthermore, by adjusting the orientation, period, depth, etc. of the sub-wavelength grating, selective transmission (reflection/absorption) could be realized, and therefore it has attracted widespread attention in many applications [23,24]. However, it is also highly desired in some fields to make the working band of photonic device regulated dynamically and reversibly. Therefore, solid-based photonic devices with invariable structural parameters exhibit severe limitations. In recent years, researchers have made great efforts to develop adjustable plasmonic devices. An effective method is to exploit the remarkable properties of plasmons—high sensitivity to surrounding dielectric materials of plasmonic nanostructures [25,26,27]. For example, Wang and Chumanov have demonstrated electrochemical modulation of the LSPR intensity and frequency in Ag nanoparticles by coating with a tungsten oxide gel [28]. Zheng et al. applied a bistable, redox-controllable rotaxane molecule to a gold nanodisk array to achieve active adjustment of plasmon resonance by chemical conversion between its oxidized and reduced states. [29].

In this paper, we present an adjustable filter based on a grating structure by creatively adopting an advanced electrowetting fluid-manipulation technology in a double-layer metal–fluid–dielectric configuration. By changing the period of the grating, we have achieved the reflection of CMY (cyan, magenta, yellow) primary colors. Furthermore, the tinge of each primary color is dynamically and reversibly regulated for the first time by simply varying the electrowetting voltage applied to change the distribution of the fluids. And due to the fast speed of electrowetting regulation, the response time of the device is around 10 ms, which is more responsive and superior than many other methods. Furthermore, the benefits also include reversible regulation, miniaturization, and so forth. This tunable filter allows for a variety of tinge choices for each kind of colors, extends the color field of the display device without the demand of increasing the number of components. In addition, our fluid-based design is reusable and flexible, compared to the solid-based devices.

## 2. Materials and Methods

Figure 1a,b shows the principle of the adopted advanced fluid manipulation technique, which is generally named as electrowetting. Electrowetting technique is used in our research to achieve contact angle modulation. The contact angle of the conducting fluid on the solid surface is determined by the force balance at the contact point. The initial equilibrium contact angle, as shown in Figure 1a, $\theta_{0}$, is given by Young’s equation:(1)$$\gamma_{s1}+\sigma_{12}\cos\theta_{0}=\gamma_{s2}$$

Here $\gamma_{s1}$ is the surface energy per unit area between the solid surface and conducting fluid, $\gamma_{s2}$ is the surface energy per unit area between the solid surface and insulating fluid, and $\sigma_{12}$ is the surface tension at the interface between the two fluids.

In electrowetting devices, a voltage is usually applied between the conducting fluid and the driving electrode. In many applications, a thin hydrophobic dielectric layer is deposited onto the driving electrode, which is often referred as “electrowetting on dielectric” (EWOD). In this case, the capacitance of the dielectric layer is superior to the electrical double layer capacitance at the electrode-liquid interface. By increasing the voltage, the energy stored in the capacitor get increased, and meanwhile, the effective surface energy gets reduced. Young’s equation is therefore modified as follows:(2)$$\gamma_{s1}-\frac{\varepsilon V^{2}}{2d_{f}}+\sigma_{12}\cos\theta_{ew}=\gamma_{s2}$$

Here ε is the permittivity of the dielectric, V is the applied voltage-difference, and *d_f_* is the dielectric thickness. Combining Equations (1) and (2) yields
(3)$$\cos\theta_{ew}=\cos\theta_{0}+\frac{\varepsilon V^{2}}{2\sigma_{12}d_{f}}$$

Thus, electrowetting can be well used to dynamically modify the force balance at the contact point, and thereby the contact angle gets changed accordingly, as shown in Figure 1b.

Based on the principle of electrowetting introduced above, we have established a two-layer metal grating structure model as shown in Figure 1c. In such design, silver and silicon are selected as the metal and dielectric materials respectively, forming the core architecture of grating filter. Compared with single-layer metal grating, the two-layer sub-wavelength metal grating has superior polarization and filtering performance. To integrate the electrowetting features in the model, a thin, hydrophobic polymethyl methacrylate (PMMA) layer is added onto the sidewalls of upper silver strips. Meanwhile, the upper and lower silver strips are acting as the driving and grounding electrodes respectively. A certain proportion of water and oil, acting as the conducting fluid and insulating fluid respectively, are then added to the space between the upper silver strips of the adjacent gratings. The interface curvature between the oil and water is initially determined by the force balance as predicted by Equation (1) and can be modulated by regulating the electrowetting voltage applied between the two electrodes as shown in Figure 1d–f.

As the surface tension of the oil is smaller than that of water, the initial state, that is, when the voltage is 0, the contact angle of water on the sidewall surface is relatively large, and the interface bulges in the direction of the oil, as shown in Figure 1d. During the process of increasing the applied voltage, the contact angle is gradually reduced, and the interface gradually flattens and then bulges toward the water, as shown in Figure 1e,f. Meanwhile, the media distribution in such device changes with the interface morphology, rendering the modulation of filtering function. If we reduce the voltage again, the centre point of oil–water interface will gradually recover to the previous height. Furthermore, the response time of electrowetting technology is particularly short, so the desired interface morphology can be quickly realized by applying a certain voltage.

To explore the relationship between the interface morphology and applying voltage, as well as the dynamical filtering function, COMSOL Multiphysics and the finite difference method are used in present study. For material selection in the simulation software FDTD, we chose oil with a refractive index of 1.7, water with a refractive index of 1.3, silicon in the material library, and silver in the palik (0–2 μm). The boundary condition is a periodic condition, and the light source is a 400–760 nm plane wave.

## 3. Results and Discussion

As can be predicted according to the electrowetting theory, the oil/water interface morphology between the upper silver strips can be regulated as desired in our model, by selectively varying the device parameters, e.g., the voltage applied, the spacing between strips, and the hydrophobic layer material type and thickness. During our investigation, we firstly settled our device period as 400 nm, with duty cycle as 0.5. We set the upper layer metal thickness as 100 nm, lower layer metal thickness as 140 nm, and Si layer thickness as 100 nm.

In a typical EWOD configuration, the same contact angle variation usually requires a higher applying voltage for a higher dielectric layer thickness, which is the reason why a very thin hydrophobic PMMA layer is used in present study for minimizing energy consumption as well as the component miniaturization. By slightly increasing the electrowetting voltage, the contact angle in our device can reduce significantly. Meanwhile, the interface morphology is synchronously changed from a protruding state to a concave state, as shown in Figure 2a–c. The distribution of two fluids get changed continuously, in another word, the surrounding media of the two-layer metal grating structure is changed during this whole process. Theoretically, the filtering function of the device will be also changed. It is well known that the working band of the filter can be adjusted by changing the grating period. For a specific period, changing the distribution of the liquid, that is, changing the refractive index of the environment media around the metal, can also affect the absorption of a specific wavelength.

Figure 2d–f is the corresponding electric field distribution diagram in this case. As can be observed, the electric field is mainly concentrated in the region between the ridge of the grating and the trench, that is, the transparent dielectric layer PMMA and the region where the fluid exists. From the images we can see that as the contact angle decreases, the coverage area proportion of oil which covers the upper metal surface is decreasing, and the electric field excited in the vicinity is also gradually weakened. A larger coverage area proportion of oil on the metal surface implies there is a greater dielectric constant of the surrounding environment, which will offset more free charge on the metal surface by the polarization charge generated in the surrounding medium. Then, the restoring force become smaller, and thereby the resonance energy of the surface plasmon becomes weakened. Therefore, in the process of increasing the voltage, the coverage area of oil gradually decreases, and the resonance wavelength continuously shifts blue. Figure 2g–i indicates the distribution of magnetic field, which is mainly concentrated in the region between the upper metal and the lower metal, that is, the transparent dielectric layer PMMA. It is noticed that the magnitude of the magnetic field enhancement is also related to the distribution of the fluids interface. Figure 2j shows the modulation of the resonant wavelength by different electrowetting voltages under the same structural parameters as mentioned in previous section. As can be observed, the wavelength varies from 579 nm to 542 nm continuously. The corresponding extinction tinge was also changed.

For a color filter based on the solid-state grating structure, the modulation in filtering function is usually realized by varying the periodic size. In current study, we also carried out the related investigation in our new designed fluid-based two-layer metal grating structure. Considering the interests of visible light, the periodic size is changed from 250 nm to 600 nm, with the increasing step of 50 nm. As can be observed in Figure 3a, the extinction spectra get adjusted significantly, to be specific, it continuously red-shifts from 462 nm to 720 nm with the periodic size increases. In another word, three primary colors, i.e., red, green and blue, can be obtained in our structures. More importantly, the extinction wavelength of each structure increases with the applying voltage, which further confirms that the invented fluid-based devices have fine-tuning characteristics for each color, including the primary one, under the role of EWOD voltage. Similar with the situation introduced in Figure 2, this novel phenomenon can be attributed to the continuous change in surrounding media of the fluid-based two-layer metal grating structure by the applying voltage.

In Figure 3b, we present a schematic diagram of the centre point height of oil–water interface as a function of voltage for different periodic sizes. The curves are all derived from the COMSOL software, where the device parameters are accurately settled. As can be clearly observed that the height decreases nonlinearly with the increasing voltage and gets more significantly change for a larger periodic size even under the same voltage variation.

The investigation results proved that the surrounding media distribution can be well controlled by simply applying appropriate voltage and thereby the tinge of each color obtained can be regulated and selected. On this basis, we further investigated the filtering function of two cases where the periods are 250 nm and 550 nm. We can notice that these two periods respectively correspond to the extinction colors of the blue band and the red band. Furthermore, as can be seen from Figure 3c,d, they both produce a blue shift as the voltage increases. The blue color changes from 505 nm to 462 nm, and the red color changes from 680 nm to 630 nm. The corresponding tinge changes are also indicated.

By realizing the three extinction colors of red, green and blue, we can get the reflected colors of CMY. Figure 4a shows a schematic diagram of a simple color filter in which three different periods can reflect three different colors. However, unlike the conventional solid-based color filters, the device proposed here has flexible adjustability. The change of the externally applied voltage led to the changes of oil/water interface curvature inside the device cavity and causes the shift of the spectral absorption peak position. The tinge of the reflected light of the device is then changed. More importantly, this kind of device can realize the correction of performance error induced by the fabrication deviation or design defect in structural parameters, which has a significant advantage in making a device with high requirements in tinge precision without the demand of high-level processing standards.

To provide a quantitative measure of the reflected color, we calculated the chromaticity coordinates of the spectrally reflected light and plotted it in the CIE 1931 xy color space, as shown in Figure 4b. Since the extinction color is blue, green and red, the light reflected from the device is its corresponding complementary color. Since the tinge of the extinction color varies widely, the range of tinge that can be achieved in the CIE 1931 xy color space is also quite wide. More importantly, different combinations of tinge can be achieved by applying different voltages to the gratings with different periods in the device. Integrating gratings of different periods in the same device results in a wider tinge range than conventional devices, which greatly increases the applications of the device. The colors that can be obtained are shown in Figure 4c, and each color can be realized directly or realized by mixing the colors obtained from different periods or different voltage conditions. The principle of subtractive color method is potentially useful to replace the ink and pigment to achieve various colors, avoiding the usage of the harmful substances contained in traditional inks and pigments, exhausting waste after complicated chemical processes and cumbersome processes. We can also choose different periods of grating structure to form different devices, or change the fluid material, dielectric layer thickness, grating duty cycle, metal layer height and other parameters to realize the desired color range, and thus satisfy different needs of various occasions. This further increases the flexibility of device usage. This adjustable color filter will have important application prospects in the fields of flat panel display, photovoltaic, biomimetic, green printing and security.

## 4. Conclusions

In summary, we have designed a new type of color filter in combination with advanced electrowetting fluid handling technology and double-layer grating structure. We fully consider the compatibility of fluid self-assembly methods with traditional micromachining methods. By changing the applied voltage between the conductive fluid and the driving electrode to change the environmental medium around the metal in the grating, the color of the extinction color can be dynamically and reversibly adjusted. The tunable device can be reused, the liquid parameters are easily changed, and the response speed is extremely fast. As such, there is a wide range of important potential applications in the fields of new flat panel display, photovoltaic, and so on.

## Figures and Tables

**Figure 1 nanomaterials-09-00070-f001:**
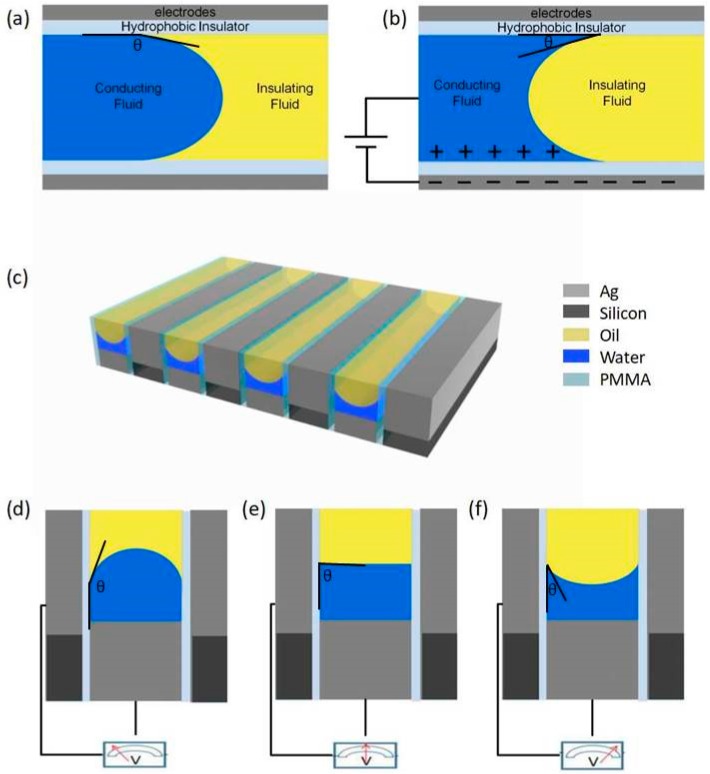
The principle of electrowetting is shown in (**a**,**b**). (**a**) The initial state and (**b**) indicates the situation after applying voltage. (**c**) The sketch of the model we created. In the figure, gray, black, yellow, blue and transparent areas respectively represent metallic silver, silicon elemental, oil, water and medium polymethyl methacrylate (PMMA). (**d**–**f**) The three different cases of the interface morphology after voltage application.

**Figure 2 nanomaterials-09-00070-f002:**
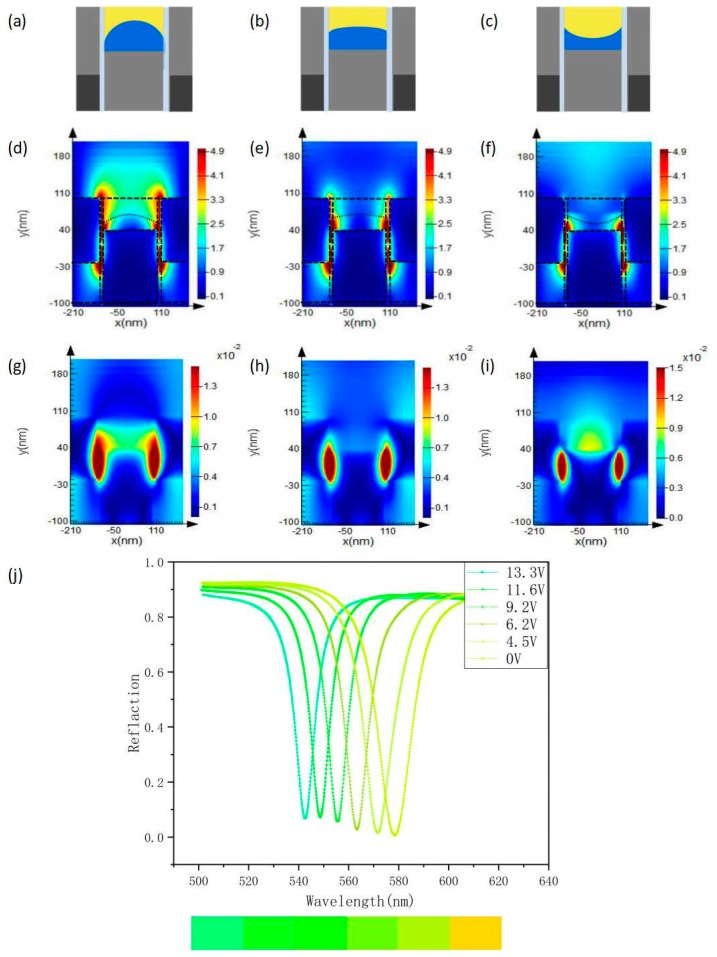
(**a**–**i**) The electric field and magnetic field distribution of the invented device with a period size of 400 nm working at the electrowetting voltage of 4 V, 8 V and 12 V, respectively. (**j**) Indicates the reflection spectrum of the device with the mentioned period size.

**Figure 3 nanomaterials-09-00070-f003:**
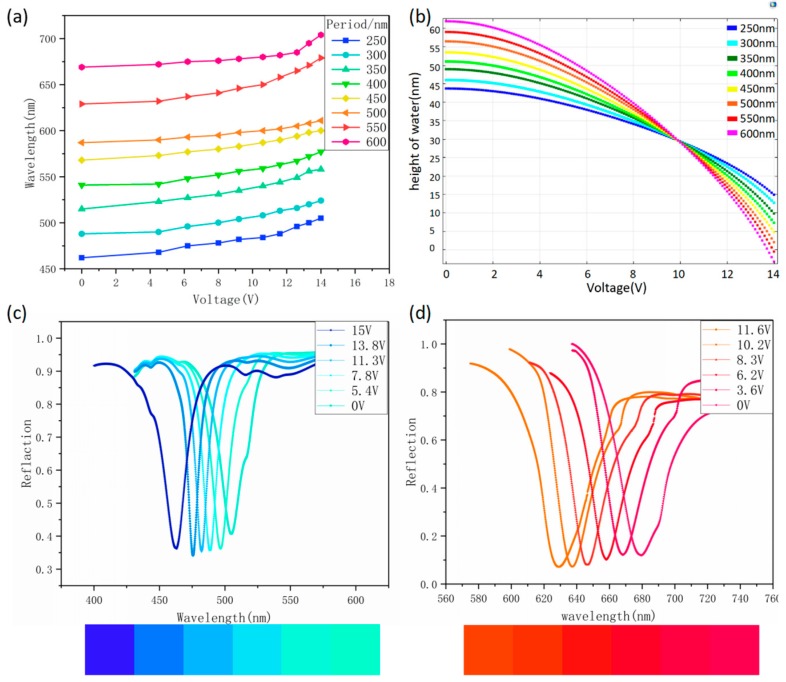
(**a**) The relationship of extinction wavelengths with the electrowetting voltages. (**b**) A schematic representation of the centre point height of the oil–water interface as a function of voltage. (**c**,**d**) Reflection peaks at different voltages for periods of 250 nm and 550 nm, respectively.

**Figure 4 nanomaterials-09-00070-f004:**
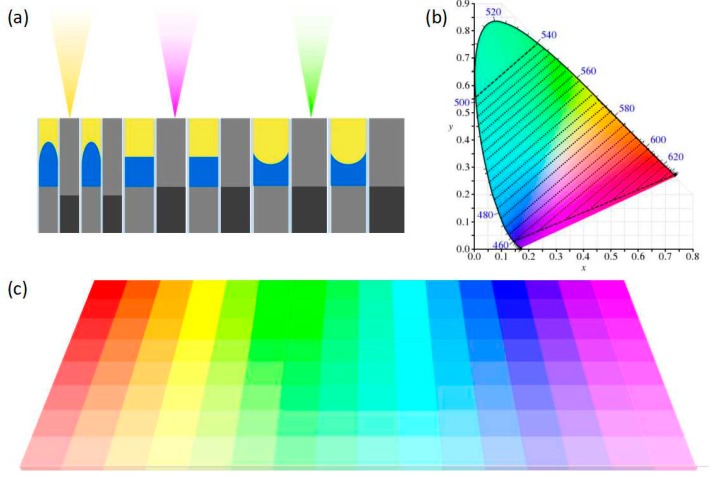
(**a**) The schematic diagram of the proposed integrated filtering device based on the grating structure. The periods of the three parts are 250 nm, 400 nm and 550 nm, respectively. (**b**) The range of gamuts that the filtering device can produce in CIE 1931 xy chromaticity coordinates (dashed line). (**c**) Color gradient map that can be achieved by selecting the appropriate parameters.

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
