# Peer review of "Dynamically Tunable Light Absorbers as Color Filters Based on Electrowetting Technology"

_nanomaterials, 2019, doi:10.3390/nano9010070_

Reviewer 1 Report

It is a good paper but authors should clearly state that all the results presented are just simulations. If the authors could make an experiment, even a simple one, showing the main principle of the paper (the light control by electrowetting of two different liquids) it could gain a lot more of interest. 

Author Response

Reviewer 1

Comments and Suggestions for Authors:

It is a good paper but authors should clearly state that all the results presented are just simulations. If the authors could make an experiment, even a simple one, showing the main principle of the paper (the light control by electrowetting of two different liquids) it could gain a lot more of interest.

Reply:

Thanks for your kind reminding. We are sorry that the misunderstanding has been caused by the unclear description in our previous manuscript. We have already explained in the abstract and body sections that the results in this manuscript are all simulations and added the instructions in simulation software and parameters we used. As for the experiments, due to the limitations of revision time of this journal and our own experimental conditions, we are temporarily unable to complete the experimental investigation. Of course, experimental verification and research are our key tasks in the next step. If we could get good experimental results, we will publish them further.

Reviewer 2 Report

The manuscript deals with the realization of an adjustable filter based on a grating structure by adopting an electro-wetting fluid-manipulation technology. However, the manuscript is poor in describing in a clear and accurate way the experimental novely of the technology. The 'materials and methods' suffer from the lack of material part. Moreover, the authors claim the invenction of a new device with new eccellent properties, without reporting a substantial and adequately discussed comparison with the standard technology in order to discuss in detail the origiinality of their product. The language used to write the article is rather abrupt and too semplicistic for a scientific publication.

Author Response

Reviewer 2

Comments and Suggestions for Authors:

The manuscript deals with the realization of an adjustable filter based on a grating structure by adopting an electro-wetting fluid-manipulation technology. However, the manuscript is poor in describing in a clear and accurate way the experimental novely of the technology. The 'materials and methods' suffer from the lack of material part. Moreover, the authors claim the invenction of a new device with new eccellent properties, without reporting a substantial and adequately discussed comparison with the standard technology in order to discuss in detail the origiinality of their product. The language used to write the article is rather abrupt and too semplicistic for a scientific publication.

Reply:

Thanks for your comments. Firstly, electrowetting is an advanced technique that uses voltage to control the distribution of fluids. Its benefits include short response times (less than 10ms), reversible regulation, miniaturization, and so on. Our innovation is to combine electrowetting technology with the properties of surface plasmons, which is sensitive to the surrounding dielectric constants. By applying voltage to regulate fluid distribution around metal nanostructures, a new type of dynamically tunable color filter can be realized. In current version, we have revised the related contents to well describe the technology novelty. The material parameters adopted in the simulation software have also been well introduced in the ‘Materials and Methods’ section. Secondly, in the introduction part of the article, we have added the description of techniques adopted by others. At present, there are some methods for dynamically regulating surface plasmon characteristics. Most solid-state devices have problems such as slow speed and irreversible adjustment. In recent years, the development of new liquid crystal technology has increased its speed, but there are still some problems such as temperature sensitivity. Comparatively speaking, we have proposed an innovative structure, which provides new ideas and inspiration for the field of color filter design. Finally, we have tried our best to modify the language and grammatical errors in the article to meet the publishing requirements.

Reviewer 3 Report

This paper propose a method to make a dynamically tunable light absorbers as color filters by using electrowetting technology of which color performances were investigated by FEM simulations. By adjusting the electrowetting voltage, the fluid distribution can be changed, resulting fine tunning of colors. Simultaneous control of voltage and the period of grating enables a large color gamut range. Although the idea that electrowetting technology can change colors is interesting, the tunable color range is quite narrow. I recommend this paper must be improved for the publication in Nanomaterials. 

Line 205-213. The tuning range of the color by adjusting voltage is very narrow. Only tinge of the color changes. Change of grating period can not be considered for the  dynamic color tunning. Because after processing the structure, the period of a structure is fixed. Therefore, the applications such as flat panel display, biomimetic, etc., may not be allowed only by using the suggested technique. Authors should suggest a method to enlarge the dynamic range of colors for the practical applications.

In fig. 4(c), please indicate the corresponding voltages and periods. 

There are a lot of researches to achieve the tunable color filters. Authors should introduce recent researches on tunable color filters.

Author Response

Reviewer 3

Comments and Suggestions for Authors:

This paper propose a method to make a dynamically tunable light absorbers as color filters by using electrowetting technology of which color performances were investigated by FEM simulations. By adjusting the electrowetting voltage, the fluid distribution can be changed, resulting fine tunning of colors. Simultaneous control of voltage and the period of grating enables a large color gamut range. Although the idea that electrowetting technology can change colors is interesting, the tunable color range is quite narrow. I recommend this paper must be improved for the publication in Nanomaterials.

Line 205-213. The tuning range of the color by adjusting voltage is very narrow. Only tinge of the color changes. Change of grating period can not be considered for the dynamic color tunning. Because after processing the structure, the period of a structure is fixed. Therefore, the applications such as flat panel display, biomimetic, etc., may not be allowed only by using the suggested technique. Authors should suggest a method to enlarge the dynamic range of colors for the practical applications.

In fig. 4(c), please indicate the corresponding voltages and periods.

There are a lot of researches to achieve the tunable color filters. Authors should introduce recent researches on tunable color filters.

Reply:

Thank you for your suggestion, this is a very valuable proposal. First of all, the regulation scope of our single device is not very large, but the dynamic range of colors can be further enlarged based on the presented results by integrating different grating structures into a single device. By doing this, the continuous regulation range can cover the whole visible band. Furthermore, the device size obtained could be much smaller than the conventional ones for the realization of the same color filter effect. Therefore, this invention can effectively improve the integration of the device and the display range of the color. The related content has been revised in current version to make the readers well understand the practical scheme of the introduced technology. At the same time, we will further study other plasmonic structures, and design suitable structures that can be combined with electrowetting technology to expand the color adjustment range of the device. As for the colors shown in Figure 4(c), each one can be realized directly or realized by mixing the colors obtained from different periods or different voltage conditions. Figure 4(c) is a visual representation of the range of colors that can be achieved in Figure 4(b). Therefore, we regret that we cannot specifically mark the corresponding value of each color in the figure. Finally, we have cited and discussed the recent progress on tunable color filters in the introduction.

Reviewer 4 Report

In the current manuscript, the authors introduced a new concept of a tunable optical structure, which allows achieving various colours by applying an external voltage. They employed the so-called electrowetting on dielectrics to dynamically change the spectra of an optical filter. This is quite interesting and deserves the publication. I just have several minor comments: i) the authors didn't discuss how fast the tuning can be done? My expectation that ON and OFF times will be quite different since the relaxation can't be speeded up. ii) as a theoretical design it's a promising one, but of course, the experimental verification is desirable. The authors should discuss the possible realisations and also some issues, which might appear during the process. iii) In the Conclusion that should replace the word "invented" with "suggested" - it's not tested yet, so, it's not invented yet either.

Author Response

Reviewer 4

Comments and Suggestions for Authors:

In the current manuscript, the authors introduced a new concept of a tunable optical structure, which allows achieving various colours by applying an external voltage. They employed the so-called electrowetting on dielectrics to dynamically change the spectra of an optical filter. This is quite interesting and deserves the publication.

I just have several minor comments: i) the authors didn't discuss how fast the tuning can be done? My expectation that ON and OFF times will be quite different since the relaxation can't be speeded up. ii) as a theoretical design it's a promising one, but of course, the experimental verification is desirable. The authors should discuss the possible realisations and also some issues, which might appear during the process. iii) In the Conclusion that should replace the word "invented" with "suggested" - it's not tested yet, so, it's not invented yet either.

Reply:

i) Thanks for your comments, this is a very good question. According to current knowledge, the ON time of electrowetting is generally less than 10 ms. As you mentioned, the ON and OFF times are quite different, but the recovery time can be appropriately reduced by technical improvement, such as increasing the smoothness of the dielectric layer to reduce the hysteresis of the liquid, as well as reducing the viscosity of the fluid and so on.

ii) Thanks for your good suggestion. We have already fully considered the achievability of the device when designing the structure. We have carefully studied and confirmed that fluid self-assembly and sidewall plating electrodes are compatible with traditional micromachining methods. However, in future experiments, how to achieve uniform distribution of fluids within a small structure might be a problem to be considered. The related discussion has already been added in the text.

iii) Thanks for your kind reminding. We have changed the inappropriate terms in the conclusion and the article.

Round  2

Reviewer 2 Report

The manuscript has been implemented and most of the queries have been answered in a satisfying way. The paper is now acceptable for publication.

Reviewer 3 Report

The revised paper well addressed all the issues raised by reviewer. I recommend this paper can be published in Nanomaterials.